# Behavioral and emotional difficulties in maltreated children: Associations with epigenetic clock changes and visual attention to social cues

Keiko Ochiai[1], Shota Nishitani[1,2,3]*, Akiko Yao[2], Daiki Hiraoka[2], Natasha Y.S. Kawata[2], Shizuka Suzuki[4,5], Takashi X. Fujisawa[1,2,3], Akemi Tomoda[1,2,3,4]*

1 Division of Developmental Higher Brain Functions, United Graduate School of Child Development, Osaka University, Kanazawa University, Hamamatsu University School of Medicine, Chiba University, and University of Fukui, Fukui, Japan, 2 Research Center for Child Mental Development, University of Fukui, Fukui, Japan, 3 Life Science Innovation Center, University of Fukui, Fukui, Japan, 4 Department of Child and Adolescent Psychological Medicine, University of Fukui Hospital, Fukui, Japan, 5 Department of Science of Human Development, Faculty of Education, Humanities and Social Sciences, University of Fukui, Fukui, Japan

* nishi.s@yamanashi.ac.jp (SN); atomoda@u-fukui.ac.jp (AT)

## Abstract

Research indicates that childhood maltreatment leads to adverse outcomes later in life and accelerated aging. However, few studies have investigated how age acceleration manifests during childhood. This study aimed to investigate the impact of child maltreatment on DNA methylation age (mAge) acceleration using a case-control study design and its association with visual attention and behavioral and emotional outcomes in maltreated children (CM). We hypothesized that CM experience atypical aging, which adversely affects their behavioral and emotional outcomes by disrupting the cognitive development necessary for forming interpersonal relationships. The study included 36 CM and 60 typically developing (TD) children with an average age of 4–5 years. We compared their DNA mAge acceleration, measured through buccal DNA samples. Additionally, we conducted a behavioral assessment of their cognitive functions related to interpersonal interactions, using an eye-tracking system to measure their gaze points at various social stimuli. Behavioral and emotional outcomes were evaluated using the Strength and Difficulties Questionnaire (SDQ). The results showed that CM exhibited significantly higher mAge acceleration and spent significantly less time gazing at the eye region during facial expression presentations. While a significant association between these attributes was observed, a comprehensive path analysis revealed that each attribute independently correlated with higher SDQ scores, suggesting that child maltreatment leads to these difficulties through accelerated aging and decreased eye contact. This study provides significant insights into how child maltreatment impacts children's development. It demonstrates that mAge acceleration and reduced attention to the eye region are critical factors associated with the adverse behavioral and emotional outcomes observed in maltreated children. These findings

**Data availability statement:** All relevant data are within the paper and its Supporting Information files.

**Funding:** All phases of this study were supported by AMED (20gk0110052, AT and SN), JSPS KAKENHI Scientific Research (A) (19H00617 and 22H00492, AT), Challenging Exploratory Research (Houga) (21K18499 to AT), Scientific Research (C) (20K02700, SN), a grant-in-aid for "Creating a Safe and Secure Living Environment in the Changing Public and Private Spheres" from the Japan Science and Technology Corporation (JST)/Research Institute of Science and Technology for Society (RISTEX), a research grant from the Strategic Budget to Realize University Missions (AT), research grants from the University of Fukui (FY 2019 and 2020 to SN), a grant-in-aid for translational research from the Life Science Innovation Center, University of Fukui (LSI20305 and LSI22202 to SN), and a grant for life cycle medicine from the Faculty of Medical Sciences, University of Fukui (SN). The funders had no role in study design, data collection and analysis, decision to publish, or preparation of the manuscript.

**Competing interests:** The authors have declared that no competing interests exist.

**Abbreviations:** ACE, adverse childhood experiences; ANOVA, analysis of variance; ASD, autism spectrum disorders; AU, arbitrary unit; CM, maltreated children; CPS, Child Protective Services; DQ, developmental intelligence quotient; FDR, False Discovery Rate; KSPD, Kyoto Scale of Psychological Development; mAge, methylation age; SDQ, Strength and Difficulties Questionnaire; VEX-R, Violence Exposure Scale for Children-Revised; WISC-IV, Wechsler Intelligence Scale for Children-Fourth Edition.

highlight the importance of early intervention and support for maltreated children to mitigate the long-term effects of accelerated aging and social cognitive deficits.

## Introduction

Child maltreatment is associated with numerous adverse health outcomes, including increased risks of death, disease, impaired immunity, cancer, myocardial infarction, and psychiatric disorders later in life [1–5]. Epidemiological studies have also reported that child maltreatment can accelerate certain aspects of aging, such as causing premature puberty [6,7], and biological markers such as telomere length have been used to measure this accelerated aging [7–15]. However, findings using telomere length are inconsistent. Therefore, it is essential to examine the effects of child maltreatment on accelerated aging using new biological metrics and evidence to clarify the adverse health and developmental outcomes caused by child maltreatment.

Epigenetic age acceleration, calculated by the deviation of DNA methylation age (mAge) from chronological age [16], has recently gained attention as a novel biomarker of aging. Some studies have shown mAge acceleration is associated with psychological trauma, such as adverse childhood experiences (ACE) [17] and PTSD in adults [18–20]. In pediatric populations, several studies have reported correlations between mAge acceleration and exposure to child maltreatment or ACE. Jovanovic *et al.* [21] measured violence exposure in children aged 6–13 years using the Violence Exposure Scale for Children-Revised (VEX-R) alongside heart rate measures. They found that mAge acceleration was associated with children's direct violence experience and decreased heart rate. Children who appeared older than their chronological age had twice as much violence exposure as others, and their heart rate resembled that of adults. Sumner *et al.* [22] assessed exposure to maltreatment, psychological trauma, and other adversities through interviews and self-reports from children aged 8–16 years and questionnaires completed by caregivers, finding that violent experiences, rather than deprivation, were associated with mAge acceleration. Tang *et al.* [23] investigated cumulative ACE scores from ages 0–14 in a large population cohort ($n = 974$) with an average age of 17 years, reporting that emotional and physical abuse were associated with mAge acceleration in girls but not in boys. However, these findings should not yet be considered conclusive for several reasons. First, although subjective evaluation is important, some studies did not rely on objective measures such as whether the maltreatment warranted intervention by Child Protective Services (CPS), which legally removed children from their parents and placed them in institutions. Instead, they obtained self-reported trauma scores from the general population and examined the relationship between adverse experiences and mAge acceleration. Second, these studies used Horvath's multi-tissue epigenetic clock, which is known to be unreliable when applied to pediatric populations [24].

Our pilot study, which employed the newly developed Pediatric-Buccal-Epigenetic (PedBE) clock [24], revealed that mAge was accelerated in maltreated children (CM)

with an average age of five years who had experienced CPS intervention, compared to age-matched children in the general population [25]. However, this finding was initially reported in a brief letter [25], which aimed to rapidly communicate the key discoveries but lacked detailed analyses and a comprehensive dataset due to space limitations. Concurrently, Dammering *et al.* also utilized the PedBE clock to examine mAge acceleration in young children with internalizing disorders, observing similar mAge acceleration in those who had experienced maltreatment [26]. These findings collectively suggest that the mAge acceleration in CM, as evaluated by PedBE, is a plausible phenomenon that warrants further conclusive investigation.

In addition to the mAge pilot study, the participants also took part in a separate behavioral study investigating social cognitive processing via eye gaze [27]. This study explored the potential effects of visual trauma in CM, which may contribute to the development of Post-Traumatic Stress Disorder (PTSD) [28]. PTSD has been associated with visual disturbances, including Post-Traumatic Vision Syndrome (PTVS) [29]. Indeed, our group has reported reduced retinal thickness in CM and its association with the visual cortex of the brain [30], as well as reduced visual cortex volume in young adults with a history of sexual abuse [31].

Furthermore, previous research suggests that CM may exhibit a tendency to avoid eye contact, which could lead to or occur alongside social anxiety [27]. Individuals with social anxiety often find eye contact aversive, reinforcing avoidance behaviors [32]. In severe trauma cases, such as child maltreatment, individuals may disengage from social interactions by staring into the distance, reflecting dysregulation in the social engagement system [33]. This pattern aligns with observations in children with autism and the polyvagal theory of trauma [34]. Moreover, people with PTSD may display distinct pupil responses, such as reduced constriction to novel stimuli or excessive dilation to emotional stimuli, reflecting heightened sensitivity to stressors [35]. Consistent with these findings, our prior behavioral study [27], conducted with the same population as the mAge pilot study [25], revealed that CM, at an average age of 5, spent significantly less time gazing at eyes during facial image presentations compared to their peers. This reduced eye-gazing time was further associated with social-emotional behavioral problems and autism spectrum disorders (ASD) [36,37], suggesting its potential utility as an objective metric to evaluate the psychosocial impact of CM.

Building on these findings, the present study extends additional samples and provides a more comprehensive analysis to deepen our understanding of mAge acceleration and eye gaze behaviors in CM. First, we obtained epigenetic data from buccal epithelial cells and conducted a case-control comparison of mAge acceleration calculated using the PedBE clock. Second, we performed a cognitive eye-tracking task to evaluate the time spent gazing at various social cues, including facial images, using an eye-tracking device (Gazefinder®). Third, we examined whether the time spent gazing at social cues, considered a behavioral indicator of maltreatment in children, was associated with mAge acceleration. Finally, we investigated whether mAge acceleration and atypical eye-gazing patterns mediated the behavioral and emotional difficulties observed in CM. Our central hypothesis was that mAge would be accelerated by exposure to maltreatment and associated with time spent gazing at social cues, behavioral and emotional difficulties, and a recorded history of maltreatment.

## Methods

### Ethics approval and consent

The study protocol was approved by the Ethics Committee of the University of Fukui (Assurance no. 20140142, 20150068, and 20190107) and conducted in accordance with the Declaration of Helsinki. Written informed consent for participation was obtained from all parents or childcare facility directors.

### Participants

A total of 120 Japanese children participated in this study. The first group, comprising 50 children (CM: 21, TD: 29), participated between April 2014 and March 2019 as part of our previous behavioral study (Experiment 1) [27]. The remaining children 70(CM: 33, TD: 37) participated between April 2020 and March 2022 (Experiment 2) and were then added to the

dataset. The pilot study for mAge acceleration [25] included 56 individuals, consisting of all participants from Experiment 1 plus six individuals from Experiment 2. The sample size for the pilot study was limited because the microarray required to calculate mAge was not completed for all Experiment 2 samples at that time. In this study, mAge data (56 individuals) from the pilot study [25] and eye-gaze data from the behavioral study [27] (50 individuals) were used for the secondary data analysis.

All participants were assessed for their intelligence quotient (IQ) using the Wechsler Intelligence Scale for Children-Fourth Edition (WISC-IV) [38] or the Tanaka Binet Intelligence Scale-Fifth Edition (Japanese version of the Stanford-Binet Test) [39]. Their developmental intelligence quotient (DQ) was assessed using the Kyoto Scale of Psychological Development (KSPD) [40], the Enjohji Developmental Test [41], or the Denver Developmental Screening Test (Denver II) [42]. Data cleaning was conducted to exclude participants with duplicates (CM: 10), no maltreatment history (CM: 1), maltreatment within the first month of life (CM: 1), not DNA (CM: 1), age under one year (CM: 4, TD: 4), and presence of ACE (TD: 2). The CM group ultimately consisted of 36 children who had experienced maltreatment, had been legally removed from the care of their biological parents by Child Protection Services (CPS), and were sheltered in residential childcare facilities. They had a history of physical or emotional abuse or neglect before coming to the facility (ICD-10-CM Code T74). The TD group consisted of 60 children, all recruited from the local community.

Detailed demographic information is presented in Table 1. The study protocol was approved by the Ethics Committee of the University of Fukui (Assurance no. 20140142, 20150068, and 20190107) and conducted in accordance with the Declaration of Helsinki. All parents or childcare facility directors provided written informed consent for participation in the study.

## Psychological and behavioral characteristics assessment

ACE were scored for both the CM and TD groups by parents or caregivers at the residential childcare facility [43]. We used a Japanese version, containing nine items modified for Japanese children [44]. The ACE score ranges from 0 to 9, representing the total number of childhood adversities (one count per type of abuse) experienced before 18 years of age. To assess social-emotional problems, parents or caregivers at the residential childcare facility completed the Strengths

**Table 1. Demographic Characteristics of Participants.**

| | CM (*n*=36) | TD (*n*=60) | Statistics | *P-value* |
|---|---|---|---|---|
| Gender (Male/ Female) (%) | 20/ 16 (55.6/ 43.4) | 31/ 29 (51.7/ 48.3) | $\chi^2(1) = 0.14$ | 0.71 |
| Age (years), *Mean (SD)* | 5.9 (2.4) | 4.5 (1.8) | $t(94) = 3.24$ | 0.002 |
| ACE total, *Mean (SD)* | 2.6 (1.6) | 0.0 (0.0) | | |
| Types of maltreatment (%)<br> Physical abuse<br> Emotional abuse<br> Neglect | 6 (16.7)<br>19 (52.8)<br>31 (86.1) | NA<br>NA<br>NA | | |
| Duration (years) of maltreatment, *Mean (SD)* | 2.2 (2.0) | NA | | |
| Duration(years) elapsed from maltreatment, *Mean (SD)* | 3.4 (2.4) | NA | | |
| IQ/ DQ[a]<br>WISC-IV(FIQ)(CM:*n*=10,TD:*n*=16)<br>KSPD (CM: *n*=24, TD: *n*=30)<br>Enjohji (CM: *n*=1)<br> Tanaka-Binet (CM: *n*=1)<br>Denver (TD: *n*=6) | 89.5 (13.9)<br>88.7 (11.8)<br>73<br>71 | 105.8 (11.4)<br>102.3 (9.9)<br>6/ 6[b] | $t(24) = 3.27$<br>$t(53) = 4.67$ | 0.003<br>< 0.0001 |
| SDQ total, *Mean (SD)*[c] | 12.29 (6.2) | 7.70 (4.3) | $t(89) = 3.94$ | 0.0002 |

[a]: No IQ/ DQ assessments were conducted on eight TD. [b]: number of individuals higher than the cut off score (80). [c]: one CM and four TD have no data. ACE: Adverse Childhood Experience, WISC-IV: Wechsler Intelligence Scale for Children - Fourth Edition, KSPD: The Kyoto Scale of Psychological Development, Enjohji: Enjohji Developmental Test, Tanaka-Binet: Tanaka Binet Intelligence Scale-Fifth Edition, Denver: Denver Developmental Screening Tests, SDQ: Strength and Difficulties Questionnaire

and Difficulties Questionnaire (SDQ), consisting of 25 items. The SDQ is widely used across different cultures and has demonstrated reliability and validity in both the original and Japanese versions [45].

## Calculation of epigenetic age acceleration

Buccal swab samples were collected from each individual, with one sample taken in Experiment 1 and four samples in Experiment 2, using a commercially available cotton swab. DNA was extracted using the QIAamp DNA Mini Kit (QIAGEN, Venlo, The Netherlands) and quantified with the Qubit™ dsDNA HS Assay Kit (Thermo Fisher Scientific Inc., Pittsburgh, PA, USA). Genomic DNA (500 ng) was processed using the Illumina® MethylationEPIC array. A quality check was conducted based on the Psychiatric Genomics Consortium-EWAS quality control pipeline [46]. Samples with probe detection call rates below 90% and an average intensity value either below 50% of the experiment-wide sample mean or less than 2,000 arbitrary units (AUs) were filtered out using CpGassoc [47]. Probes with low quality (detection $P$ values > 0.01) were marked as missing. Probes missing in more than 10% of samples within the studies were filtered out, and cross-hybridizing probes were removed. Finally, 819,669 probes passed quality control and were included in the analyses. Using these probes, we performed single-sample Noob normalization with the minfi package [48]. To eliminate any chip and positional batch effects, we applied ComBat, preserving participants' age and gender via the sva package [49]. PedBE mAge, which accurately estimates mAge in pediatric buccal cells, was calculated using the method developed by McEwen *et al.* [24]. The PedBE mAge was regressed against chronological age, and the unstandardized residuals were used to measure mAge acceleration [25].

## Gaze pattern measurement

As previously documented [27,36,37,50], we measured children's gaze patterns using Gazefinder® (JVC KENWOOD Corporation, Kanagawa, Japan), an eye-tracking system designed to record responses to visual stimuli. The experiments were conducted in a quiet room at the childcare facility for the CM group and at the university research laboratory for the TD group, conducted between 9:00 and 17:00 [27]. Children were seated approximately 70 cm in front of the eye-tracking monitor on a small chair. Calibration involved directing children to fixate on an animated animal displayed in five different locations on the monitor; recalibration occurred if calibration quality was insufficient at any point. Stimulus movies were presented in a consistent sequence to all participants, with each sequence serving as a single trial. Between stimulus movies, an attention-grabbing animation accompanied by a verbal cue ("Hey! Look!") was shown at the center of the monitor to reorient children's attention to the stimuli.

Gazefinder® stimuli included short movies categorized into four types of social cues: (a) human faces, (b) people and geometric patterns, (c) biological motion of a human, and (d) objects with or without pointing gestures. Each stimulus had three Areas of Interest (AOIs): the first AOI, labeled high social, highlighted cues with significant social salience ("Eyes" in [a], "People" in [b], "Upright figure" in [c], and "Pointed" in [d]). The second AOI, low social, represented less or non-social cues with higher salience ("Mouth" in [a], "Geometry" in [b], "Inverted figure" in [c], and "Non-pointed" in [d]). The third AOI, background, represented areas with lower salience. Additional details and snapshots of the four stimuli types can be found in our previous studies [27–29]. Gaze pattern could not be measured for three TD participants, who were subsequently excluded from gaze data analyses. No significant group differences were observed in mean gaze rate (mean ± SD; CM: 0.939 ± 0.046, TD: 0.925 ± 0.060, *t*-test: $P = 0.25$).

## Statistical analysis

**Sample size calculation.** Before collecting additional data for Experiment 2, the total sample size (Experiment 1 and Experiment 2) of the behavioral data was calculated using the formula for independent two-sample t-tests. Assuming a medium effect size (Cohen's d = 0.5), an alpha of 0.05, and a power of 0.80, the calculation indicated that a total sample size of 106 (CM: 42, TD: 64, Allocation ratio TD/CM = 1.5) would be necessary to detect a significant difference.

**Group comparison of mAge acceleration.** To examine the impact of group on mAge acceleration, we conducted a two-tailed *t*-test to compare mAge acceleration between groups. Additionally, we performed multiple linear regression analysis, using group as an independent variable adjusted for age and gender, to predict mAge acceleration.

Variables and measurement

Dependent variable: mAge acceleration

Independent variable: Group (CM/TD)

Confounding variables: Age and gender (multiple linear regression analysis)

**Group and experimental condition comparison of gaze fixation.** To assess the influence of the group on each type of social cue (face, people, motion, pointing), we calculated the percentage of gaze fixation on three AOIs and performed a two-way mixed analysis of variance (ANOVA). This analysis used AOIs as dependent variables, with high social, low social, and background AOIs as within-subjects variables and group (CM or TD) as a between-group variable. We applied the Benjamini-Hochberg False Discovery Rate correction for multiple testing.

Variables and measurement

Dependent variable: The percentage of gaze fixation

Within-subjects variables: AOIs (high social, low social, and background)

Between-group variables: Group (CM/TD)

**Association analysis between mAge acceleration and gaze fixation in CM.** When identifying atypical patterns of visual attention to social cues in CM, Pearson correlation analyses examined associations between mAge acceleration and these gaze patterns. Furthermore, multiple linear regression analysis was used, with atypical gaze patterns as independent variables adjusted for age and gender, to predict mAge acceleration.

Variables and measurement

Dependent variable: mAge acceleration

Independent variable: The percentage of gaze fixation

Confounding variables: Age and gender (multiple linear regression analysis)

## Sensitive analysis for maltreatment history on mAge acceleration, gaze fixation, and emotional difficulties in CM

In addition, we explored the relationship between mAge acceleration, atypical gaze patterns, behavioral and emotional difficulties, and maltreatment history. For a detailed description of the statistical methodology, see Fujisawa TX et al. (2018) [30].

## Variables and measurement

Dependent variable: mAge acceleration, the percentage of gaze fixation, and SDQ total scores
Independent variable: Maltreatment history (maltreatment type, number of types experienced, maltreatment duration, and time elapsed since intervention)

**Serial mediation analysis.** We conducted serial mediation analysis to assess whether mAge acceleration and atypical visual attention patterns mediated the association between child maltreatment and SDQ total scores with adjusted for age and gender (Residues were calculated for the above dependent, independent, and mediating variables using confounding factors). Indirect effects were tested using bootstrap resampling (2,000 samples) for confidence intervals, using the lavaan

package [51] in R statistical software (version 4.2.1) [52]. Bootstrapping effectively addresses the challenges of small sample sizes, making it an essential method for robust mediation analysis [53]. Statistical significance was set at $P < 0.05$ for all analyses. Five participants (CM: 1, TD: 4) without SDQ data and three (TD: 3) without eye gaze data were excluded from the analysis. The final sample with no missing data consisted of 35 CM and 53 TD.

Variables and measurement

Dependent variable: SDQ total scores

Independent variable: Group (CM/TD)

Mediators: mAge acceleration and the percentage of gaze fixation

Confounding variables: Age and gender

## Results

Types of maltreatment (%): CM participants may experience multiple types of maltreatment, which is why the total does not match the sample size.

### Between-group comparisons of mAge acceleration

mAge acceleration was significantly advanced in CM compared to TD ($t = 2.02$, $df = 94$, $P = 0.046$; see Fig 1). The effect size was medium, with a Cohen's d of 0.42. This significance persisted when applying a multiple linear regression model (CM: $\beta = 0.51$, $t = 2.35$, $P = 0.02$, Age: $\beta = -0.13$, $t = -1.23$, $P = 0.22$, Gender: $\beta = 0.22$, $t = 1.08$, $P = 0.29$).

### Between-group comparisons of gaze fixation

A two-way mixed ($2 \times 3$) ANOVA was performed to examine differences in gaze fixation percentages between CM and TD groups across different AOIs. For stimuli depicting "human faces," there was no main effect of group ($F(1, 91) = 2.54$, $P = 0.12$, $\eta G^2 = 0.003$), but a significant interaction between group and AOIs was observed ($F(2, 182) = 6.13$, $P = 3.00e\text{-}03$, $\eta G2 = 0.06$). Post-hoc analyses revealed significant group differences in all AOIs: "eyes" (high social) ($F(1, 91) = 4.20$, $P = 0.043$, $P_{adj} = 0.043$, $\eta G2 = 0.04$), "mouth" (low social) ($F(1, 91) = 8.03$, $P = 0.006$, $P_{adj} = 0.018$, $\eta G2 = 0.08$) and "other" (background) ($F(1, 91) = 4.75$, $P = 0.03$, $P_{adj} = 0.043$, $\eta G2 = 0.05$) (see Fig 2a). Conversely, for the remaining types of social cues ("people and geometry," "biological motion," and "finger pointing"), there were no main effects of group on gaze fixation percentage ($F(1, 91) = 1.82$, $P = 0.18$, $\eta G2 = 0.001$; $F(1, 91) = 0.15$, $P = 0.70$, $\eta G2 = 4.81e\text{-}05$; $F(1, 91) = 0.84$, $P = 0.36$, $\eta G2 = 0.001$, respectively) and no significant group×AOI interactions ($F(2, 182) = 2.66$, $P = 0.07$, $\eta G2 = 0.03$; $F(2, 182) = 0.17$, $P = 0.84$, $\eta G2 = 0.002$; $F(2, 182) = 0.31$, $P = 0.74$, $\eta G2 = 0.003$, respectively) (see Figs 2b, 2c, and 2d). These findings indicate that the CM group demonstrates atypical visual attention patterns specifically toward human faces.

Associations between mAge acceleration, gaze fixation, behavioral and emotional difficulties, and maltreatment history
mAge acceleration showed a significant association with visual attention to the eyes ($r = -0.21$, $P = 0.047$; see Fig 3a), which remained significant in multiple regression analysis (visual attention to the eyes: $\beta = -0.21$, $t = -2.03$, $P = 0.045$, Age: $\beta = -0.05$, $t = -0.45$, $P = 0.65$, Gender: $\beta = 0.22$, $t = 1.08$, $P = 0.29$). No significant associations were found for other gaze fixation areas (mouth: $r = 0.15$, $P = 0.16$, other: $r = -0.04$, $P = 0.67$). Both mAge acceleration and visual attention to the eyes were significantly associated with SDQ total score ($r = 0.24$, $P = 0.02$; $r = -0.30$, $P = 0.004$; see Fig 3b). There were no significant associations between mAge acceleration or visual attention to the eyes and maltreatment duration or time elapsed since intervention (Fig 4a depicts maltreatment history for each case). However, significant associations were found between mAge acceleration and maltreatment history attributes, such as type and duration of maltreatment. One-way ANOVA revealed significant differences in SDQ total scores among CM exposed to different numbers of maltreatment

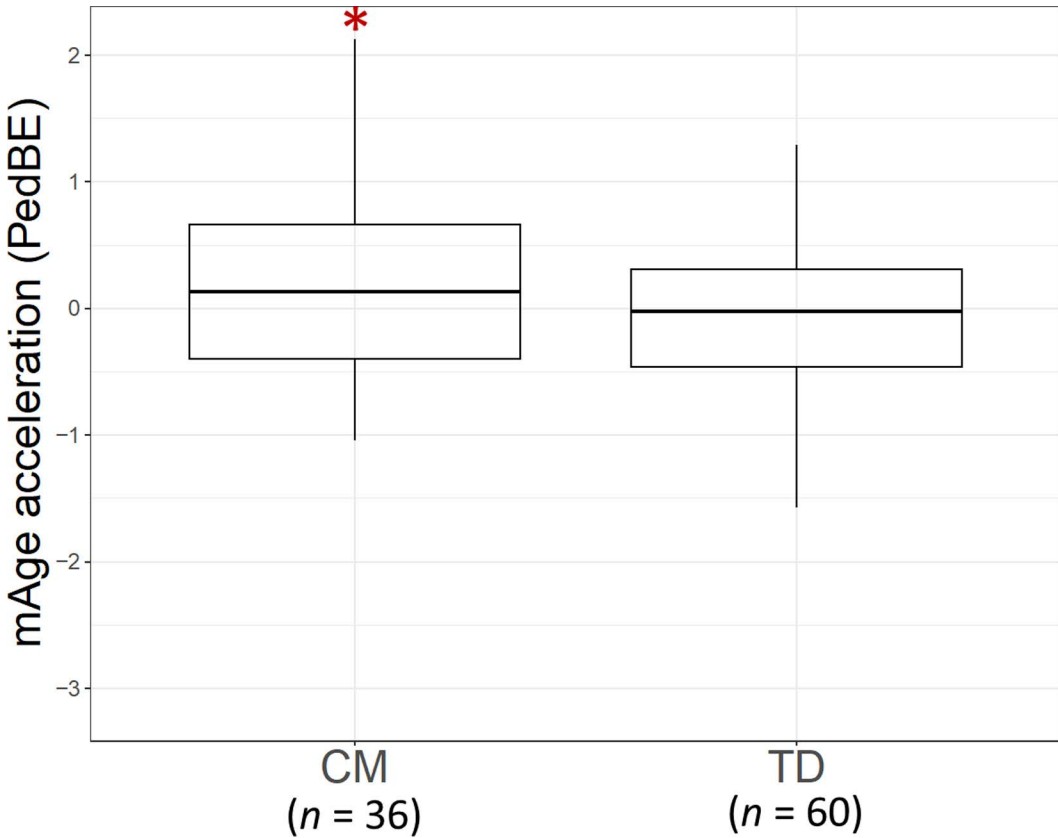

**Fig 1. Comparison of mAge Acceleration Between Maltreated Children (CM) and Typically Developing (TD) Children (*t-test: $P<0.05$).**

types ($F(2, 32) = 3.67$, $P=0.04$, $\eta G^2=0.19$) (Fig 4b). Post-hoc analysis using Tukey's honest significance test (HSD) revealed that CM exposed to three types of maltreatment had significantly higher SDQ total scores than those exposed to one type ($P=0.03$) and showed a trend toward higher scores than those exposed to two types ($P=0.08$).

## Path model

We constructed a serial mediation model (Fig 5) with mAge acceleration and/or visual attention to the eyes as mediating variables, group (CM/TD) as the explanatory variable, and SDQ total scores as the outcome variable. In this model, the path from group to mAge acceleration approached significance ($a_1=0.35$, $SE=0.18$, $P=0.056$), as did the path from mAge to visual attention to the eyes ($a_2=-0.06$, $SE=0.03$, $P=0.042$). The paths to SDQ total scores from mAge acceleration and visual attention to the eyes also showed a tendency toward significance ($b_1=1.23$, $SE=0.71$, $P=0.082$; $b_2=-8.08$, $SE=4.33$, $P=0.062$), along with a direct path from group to SDQ total scores [$c=4.566$, $SE=1.13$, $P=5.12e\text{-}05$, and $c'=3.60$, $SE=1.11$, $P=1.21e\text{-}03$]. No significant association was observed in the path from mAge acceleration to visual attention to the eyes ($d_{21}=-0.03$, $SE=0.02$, $P=0.26$). Significant indirect effects were also found [$a_1 b_1=0.43$, $SE=0.36$, 95% CI = (0.003, 1.51); $a_2 b_2=0.46$, $SE=0.36$, 95% CI = (0.01, 1.50)], indicating that both mAge acceleration and visual attention to the eyes had indirect effects on SDQ total scores. However, no significant indirect association was observed in the path from group to SDQ total scores when mediated by mAge acceleration and visual attention to the eyes [$a_1 d_{21} b_1=0.077$, $SE=0.10$, 95% CI = (-0.01, 0.55)].

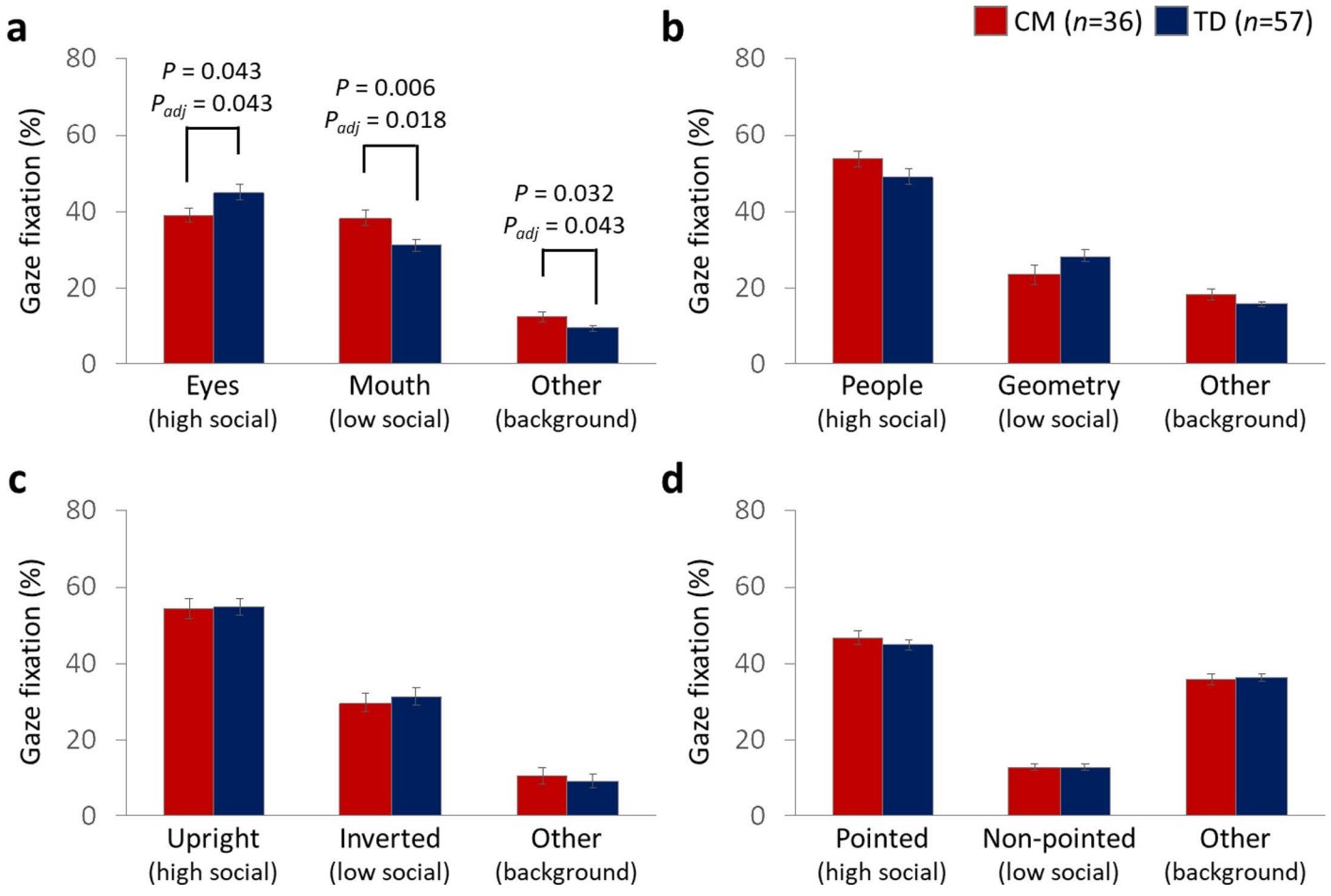

**Fig 2. Percentage of Gaze Fixation for CM and TD Groups on the Following AOIs: (a) Human Face, (b) People and Geometry, (c) Biological Motion, and (d) Finger Pointing.** The vertical axis indicates the percentage of gaze fixation on each type of social cue. AOI, Area of Interest.

Next, the initial path model was a saturated model with zero degrees of freedom, which made it impossible to calculate model fit indices, we then removed the non-significant path mAge acceleration to visual attention to the eyes ($d_{21}$) and refined the model. The results showed that the model fit the data well ($X^2$[1] = 2.001, $P$=0.157, SRMR=0.047, CFI=0.964, AIC=629.853), with the trends of significance levels of all path coefficients and indirect effects remaining preserved. Thus, these findings suggest that mAge acceleration and visual attention to the eyes are each parallel mediators of the relationship between group and SDQ total scores.

## Discussion

The present study, utilizing a larger sample size, confirmed that mAge acceleration was evident in CM (Fig 1), and these children also exhibited reduced visual attention to the eyes during facial image stimuli in CM compared to TD (Fig 2). These findings align with our previous research efforts [25,27]. While prior studies did not explore the association between these metrics, our study identified a potential link (Fig 3a), suggesting that greater atypicality in each metric correlated with higher SDQ scores (Fig 3b), indicating behavioral and emotional difficulties. However, the comprehensive path analysis did not establish a direct sequential association between mAge acceleration, visual attention to the eyes,

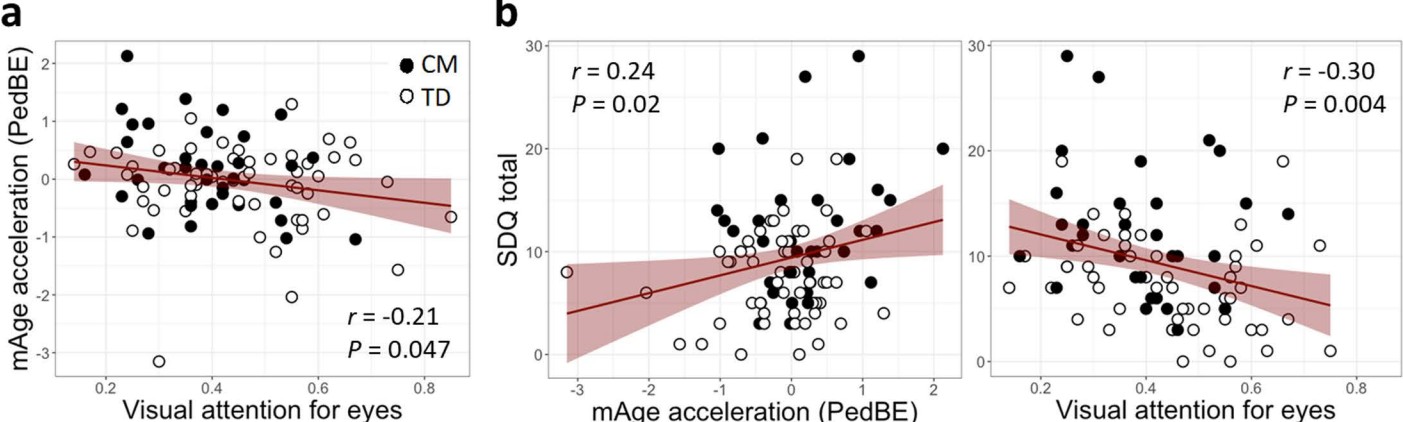

**Fig 3. (a) Associations Between mAge Acceleration and Visual Attention to the Eyes. (b) Associations Between Total SDQ Scores and mAge Acceleration (left) and Visual Attention to the Eyes (right).**

and SDQ scores (Fig 5). Instead, each metric independently correlated with higher SDQ scores. Thus, it appears that while mAge acceleration occurs in CM, it may not operate in a sequential cascade; rather, the atypical visual attention patterns and mAge acceleration might co-occur independently, both contributing to heightened behavioral and emotional difficulties in CM. These findings underscore the potential clinical implications of both mAge acceleration and diminished visual attention to social cues in understanding the social maladaptation observed in children with a history of maltreatment.

Several studies examining individuals who have experienced childhood maltreatment, adults, and patients with PTSD have reported that mAge acceleration can result from traumatic experiences [54,55]. However, most of these studies have categorized the general population based on high versus low scores on measures such as the ACE questionnaire, childhood trauma questionnaire (CTQ) [56], and similar self-reporting tools without objectively confirming severe child maltreatment experiences. Few studies have specifically focused on formally recognized cases of CM, as analyzed in our dataset.

Our pilot study, involving 25 CM and 31 TD children, initially demonstrated accelerated mAge in CM compared to TD [25]. The present study reinforces this finding with a larger cohort (36 CM and 60 TD), adding robustness to this significant finding, despite the modest effect size. However, considering the cumulative developmental impact over time, even subtle differences in mAge acceleration observed in CM may contribute to early onset of puberty [6,7] and other psychosocial challenges. In addition, our analysis utilized 94 CpG sites to predict mAge, potentially encompassing markers sensitive to severe childhood stresses and traumatic experiences such as child maltreatment. These 94 CpG sites are located on 65 genes, which were initially chosen for their strong correlation with age in the development of PeDBE clock. However, it remains unclear whether these CpG sites are associated with other factors, such as healthy aging and longevity [57] (PMID: 30241605). Recent studies have reevaluated the epigenetic clock, including PeDBE and its associated CpG sites, to determine whether they are linked to life span and overall health. Notably, for two CpG sites in PeDBE—cg04221461 (*AKT3* gene) and cg19381811 (*UBA7* gene)—associations with frailty index and Aging-GIP1 and overall health rating, respectively, have been reported [58] (PMID: 38243142). While there are currently no reports explicitly linking these to child maltreatment, it is conceivable that these sites, which were not merely correlated with age, might reflect the health status, thereby suggesting an epigenetic connection to child maltreatment.

The visual attention patterns observed in CM were primarily atypical when focusing on a person's face, as previously reported in our earlier study [27], which compared 21 CM and 29 TD children and found that CM showed reduced visual

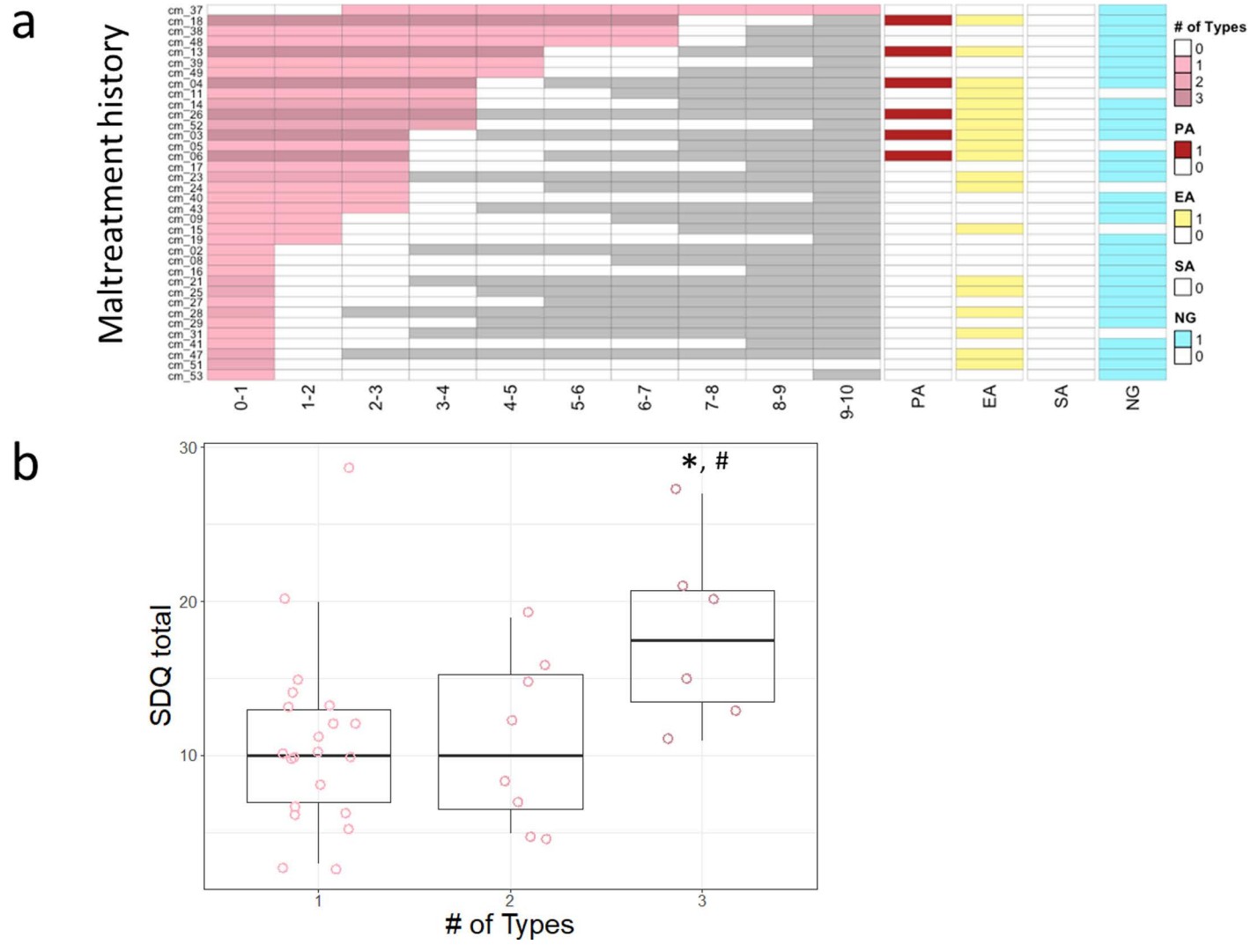

**Fig 4. (a) Structure of Maltreatment History for Each Individual.** A gray block indicates the child had not reached that age at the time of the study. PA, physical abuse; EA, emotional abuse; SA, sexual abuse; NG, neglect. **(b) Comparison of Total SDQ Scores Based on the Number of Types of Maltreatment Experienced (# of Types) in Maltreated Children (CM).** *: $P < 0.05$ (3 vs. 1), #: $P < 0.10$ (3 vs. 2).

attention to the eyes compared to TD. In the present study, this pattern was reaffirmed with a larger sample size (36 CM and 57 TD). Differences between CM and TD were observed not only in the time visual spent looking at the eyes but also in the mouth and other areas of facial images. However, these differences were likely indirectly influenced by the decreased attention CM paid specifically to the eyes. While our findings suggest a tendency for attention to shift toward the mouth rather than the eyes, there was no specific association found with psychological abuse (e.g., verbal abuse) in our results. Therefore, our focus remained on the atypical visual attention directed toward the eyes, a trait commonly observed in children with ASD [29]. This reduced gaze at the eyes is often interpreted as reflecting lower empathy and suggests a greater need for support in developing sociability and interpersonal relationships compared to TD children. Our previous study indicated that lower levels of salivary oxytocin mediated this reduced eye gaze and correlated with higher social-emotional difficulties as assessed by the SDQ subscale [27]. This underscores how lower oxytocin levels may

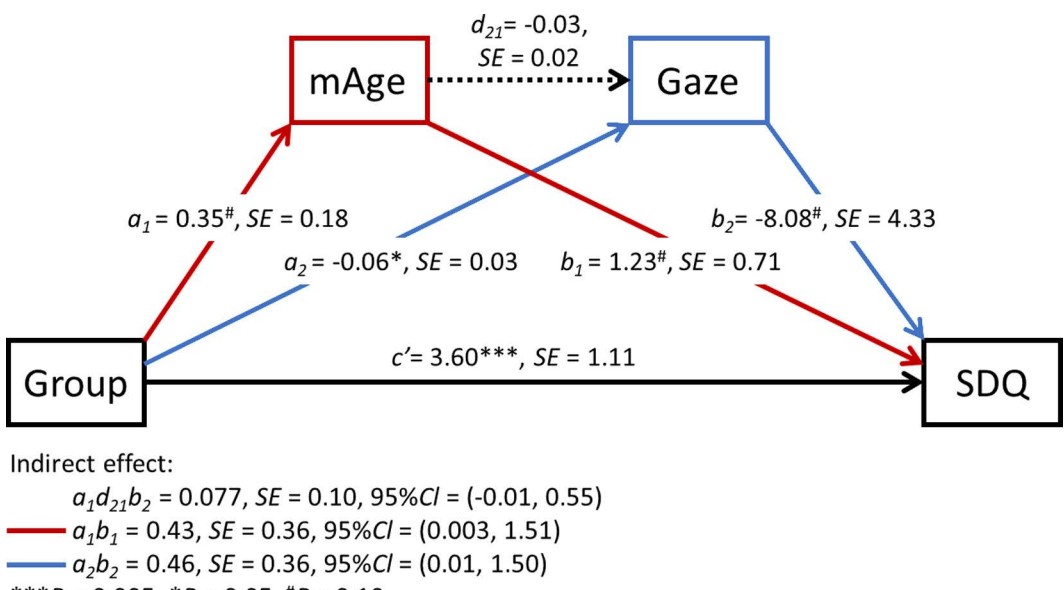

Indirect effect:
$$a_1 d_{21} b_2 = 0.077, SE = 0.10, 95\% CI = (-0.01, 0.55)$$
$$a_1 b_1 = 0.43, SE = 0.36, 95\% CI = (0.003, 1.51)$$
$$a_2 b_2 = 0.46, SE = 0.36, 95\% CI = (0.01, 1.50)$$
$***P < 0.005$, $*P < 0.05$, $^{\#}P < 0.10$

**Fig 5. Path Model from Group to mAge Acceleration and from Visual Attention to the Eyes to Total SDQ Scores.**

signify challenges in attachment relationships and vulnerability in forming interpersonal bonds. Given that these relationships form the basis of social interactions, it is plausible that CM face difficulties in forming such bonds, contributing to their reduced visual attention to the eyes.

Investigating the psychosocial phenotypes associated with mAge acceleration observed in CM is crucial to better elucidate its negative impact on them. Therefore, we examined whether there were correlations between reduced gaze time on the eyes, behavioral and emotional difficulties, and maltreatment history. Our findings indicated that less gaze time on the eyes was associated with increased mAge acceleration (Fig 3a), whereas no such associations were found for attention to the mouth or other facial areas. In addition, higher mAge acceleration and reduced gaze time on the eyes were associated with higher SDQ total scores (Fig 3b), suggesting interconnectedness among mAge acceleration, gaze patterns, and behavioral and emotional challenges. However, in our path analysis, we did not find evidence of an indirect pathway where mAge acceleration influenced gaze time on the eyes, thereby resulting in higher SDQ scores (Fig 5). Therefore, it is reasonable to interpret our results as indicating that both mAge acceleration and reduced gaze time on the eyes are influenced independently by childhood maltreatment and are individually associated with higher SDQ scores.

To interpret the separate effects of each metric on higher SDQ scores, it is important to consider why increased mAge acceleration is associated with higher SDQ scores. mAge is derived from epigenetic changes in a subset of the genome, and previous studies have reported epigenetic alterations in various genes among CM [59–64]. These changes could affect social behaviors and accelerate aging processes, potentially contributing to higher SDQ scores. Similarly, understanding why reduced gaze time on the eyes correlates with higher SDQ scores is essential. CM may have limited experience with eye contact during interactions, hindering the development of social skills over time. Indeed, adults with major depressive disorder and a history of childhood maltreatment often avoid looking at negative facial expressions conveying anger or sadness [65]. They may also exhibit other atypical visual attention patterns, similar to those observed in our research with children [27]. However, the current study suggests that the ability to understand facial expressions depends significantly on the environment in which individuals are raised. We found

that CM raised in unstable environments tend to be hypersensitive to facial expressions, whereas this hypersensitivity diminishes in more stable environments [66]. Therefore, improving the environment for CM could potentially enhance outcomes related to their social and emotional development.

This study investigated whether the type, duration, and post-intervention period of maltreatment were associated with the metrics of mAge acceleration and time spent gazing at the eyes, but no significant associations were found (Fig 4). This may be because factors common across all types of maltreatment may influence mAge acceleration and reduced eye gaze time, rather than specific types of maltreatment. All forms of maltreatment might similarly disrupt attachment formation between parents and children. Moreover, no significant association was found between mAge acceleration or reduced eye gaze time spent and the number of maltreatment types or the severity of maltreatment (Fig 4). Nonetheless, more severe maltreatment did appear to have a greater adverse impact on psychosocial outcomes, as indicated by higher SDQ total scores associated with a greater number of maltreatment types.

Although the present study reveals important findings, it has several limitations. As noted, few studies have focused on officially recognized cases of child maltreatment, and some might argue that these cases may not represent all instances of child maltreatment. However, we believe that a case-control study design was essential, as many clinical studies, including those on psychiatric and physical conditions, traditionally use this approach. Nevertheless, incorporating research designs beyond case-control studies, such as those aligned with the Research Domain Criteria (RDoC), may provide additional insights and facilitate more robust interpretations of the findings. One important limitation of this study is its cross-sectional design, which precludes any inference of causation between child maltreatment and the dependent variables, including mAge acceleration and gaze patterns. While the findings suggest significant associations, they should be interpreted as correlations rather than causal relationships, and longitudinal studies are needed to establish temporal relationships and causality. Despite adjustments in the statistical analyses, an age difference was observed between the CM and TD groups in this dataset. However, the analysis focused on mAge acceleration rather than mAge itself, which correlates with chronological age. While this approach reduces the influence of age differences, ideally, there should have been no age disparity between the groups to strengthen the validity of the findings. Finally, although we performed serial mediation analysis to test the mediation effects between variables, the sample size in this study was relatively small, even though it met the minimum goal of rule of thumb [67], and therefore we used bootstrap modeling, although larger sampling may be required for more accurate model testing.

## Conclusion

The present study, with its larger sample size, successfully replicated our previous findings on the associations between age acceleration, visual attention, and child maltreatment. It supports a model where child maltreatment contributes to behavioral and emotional difficulties through endophenotypic changes, specifically mAge acceleration and decreased attention to the eyes.

## Supporting information

**S1 Table. Final dataset.**
(XLSX)

## Acknowledgments

We would like to thank all the participants and the staff at the Research Center for Child Mental Development and Adolescent Psychological Medicine, University of Fukui Hospital, and Ms. Madoka Umemoto for their clerical support.

## Author contributions

**Conceptualization:** Keiko Ochiai, Shota Nishitani, Daiki Hiraoka, Natasha YS Kawata, Shizuka Suzuki, Takashi X Fujisawa, Akemi Tomoda.

**Data curation:** Keiko Ochiai, Shota Nishitani, Akiko Yao, Natasha YS Kawata, Shizuka Suzuki, Akemi Tomoda.

**Formal analysis:** Keiko Ochiai, Shota Nishitani, Akiko Yao, Akemi Tomoda.

**Funding acquisition:** Shota Nishitani, Akemi Tomoda.

**Investigation:** Keiko Ochiai, Akiko Yao.

**Methodology:** Keiko Ochiai, Shota Nishitani.

**Project administration:** Keiko Ochiai.

**Supervision:** Shota Nishitani, Akiko Yao, Daiki Hiraoka, Natasha YS Kawata, Shizuka Suzuki, Takashi X Fujisawa, Akemi Tomoda.

**Validation:** Akiko Yao.

**Writing – original draft:** Keiko Ochiai, Shota Nishitani.

**Writing – review & editing:** Akiko Yao, Daiki Hiraoka, Natasha YS Kawata, Shizuka Suzuki, Takashi X Fujisawa, Akemi Tomoda.

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
