## [Decision Letter · Decision Letter 0]

20 Nov 2024

PONE-D-24-33599Behavioral and emotional diffculties in maltreated children: Associations with epigenetic clock changes and visual attention to social cuesPLOS ONE

Dear Dr. Tomoda,

Thank you for submitting your manuscript to PLOS ONE. After careful consideration, we feel that it has merit but does not fully meet PLOS ONE’s publication criteria as it currently stands. Therefore, we invite you to submit a revised version of the manuscript that addresses the points raised during the review process.

**ACADEMIC EDITOR: Dear authors. Your manuscript has been reviewed favorably due to its novelty and addition to the literature, but there are some issues that need to be addressed. In particular, the study design needs to be explicit (is it a case control study or cross-sectional?) and justified in the methodology. The manuscript would also benefit from an improved variable definition, particularly for independent variables. Some other aspect just need to be clarified. Although conclusions are presented in an appropriate fashion and are supported by the data, after revision of the methodology, you may decide to update the conclusions accordingly.**==============================

We look forward to receiving your revised manuscript.

Kind regards,

Abraham Salinas-Miranda, MD, PhD

Academic Editor

PLOS ONE

Journal Requirements:

https://www.nature.com/articles/s41398-022-02159-7

In your revision ensure you cite all your sources (including your own works), and quote or rephrase any duplicated text outside the methods section. Further consideration is dependent on these concerns being addressed.

4. We note that your Data Availability Statement is currently as follows: “All relevant data are within the manuscript and in Supporting Information files.”

Additional Editor Comments:

Dear authors: The paper presents a very interesting and important topic, which examines the relationship between epigenetic aging and gaze patterns among children exposed to maltreatment compared to typically developing (non-maltreated) children. The reviewers pointed to issues in the description of the methodology that need to be addressed. Please revise the manuscript and submit your revisions point by point. Thanks.

Reviewers' comments:

Reviewer's Responses to Questions

**Comments to the Author**

1. Is the manuscript technically sound, and do the data support the conclusions?

Reviewer #1: Partly

Reviewer #2: Partly

2. Has the statistical analysis been performed appropriately and rigorously? 

Reviewer #1: Yes

Reviewer #2: Yes

3. Have the authors made all data underlying the findings in their manuscript fully available?

Reviewer #1: Yes

Reviewer #2: Yes

4. Is the manuscript presented in an intelligible fashion and written in standard English?

Reviewer #1: Yes

Reviewer #2: Yes

5. Review Comments to the Author

Reviewer #1: The manuscript by Ochiai et al. expanded on their initial pilot study of biological age in maltreated children by incorporating a larger sample size, which allowed them to do more thorough analyses. Importantly, they were able to replicate their previous findings, which concluded that biological age is accelerated in maltreated children compared to typically developing children. Although the major results presented here are not novel and have been previously reported elsewhere, I commend the authors for undertaking this study and replicating their prior work. I believe the following points should be addressed before this manuscript can be accepted for publication.

Major

1. I believe there is a typo in the title: “diffculties” should be “difficulties.”

2. I found the methods section, particularly the “Participants” subsection, a bit confusing. The authors mention on line 102 that the first group comprised 47 children, but the numbers of CM and TD in the parentheses do not add up. This also applies to the “remaining 65 children” mentioned in line 104. Could the authors please clarify?

3. It appears from Table 1 that CM participants were statistically significantly older, chronologically, than the TD group used as controls. Given that chronological age is correlated with biological age, wouldn’t it be expected that older CM participants would exhibit higher biological age? Could the authors clarify how this was addressed in subsequent analyses?

4. In the discussion (lines 355-357), the authors state that the number of maltreatment types/severity was not associated with mAge. Could they expand on this? Was this assessed in another path analysis, and can this data be shown?

5. Given the mechanistic nature of this study, it would be beneficial if the authors further discussed the potential biological implications of mAge in the discussion (PMID: 30241605). Currently, they mention that epigenetic changes have been reported in various genes among CM participants and provide references. However, it would be useful if they discussed one or some of the genes/CpG regions assessed in their mAge model and perhaps speculate on some functional relevance to CM.

Minor

1. In Table 1, I suggest providing values for each level of the categorical variables presented. For example, instead of providing data for only male participants, recode the variable as gender and provide information for all genders assessed in the study. Also, please clarify the parentheses for “ACE total.” The footnote for this table might benefit from a note clarifying that CM participants can be polyvictimized, which is why the numbers do not add up to the sample size. Additionally, consider removing the row with information about sexual abuse since it is entirely NA.

2. Consider showing individual data points for Figure 1 and adding p-values to all figures for consistency.

3. For Figure 3, consider showing additional trend lines—one specific for CM and one for TD.

4. It might benefit readers if the authors referenced specific figures in the discussion. Given some mixed results between path analysis and simple correlations (e.g., mAge correlating with gaze pattern), this could provide clearer context for readers.

Reviewer #2: I was asked to review the article entitled “Behavioral and emotional difficulties in maltreated children: Associations with epigenetic clock changes and visual attention to social cues”, submitted to PLOS ONE. The article examines an interesting topic, which is accelerated aging among individuals exposed to child maltreatment. There is currently a research gap (i.e., evidence about telomere length is inconsistent). However, there are some issues that need to be revised before this article can be accepted for publication. Details of my assessment is below.

There is a missing “i” in the word difficulties in the title.

Title: Behavioral and emotional diffculties in maltreated children: Associations with epigenetic clock changes and visual attention to social cues

Abstract: The study design is missing in the abstract and title. This must be explicit in the title or abstract for bibliographic searches and systematic reviews. It’s recommended the authors examine the STROBE guidelines for observational studies https://www.strobe-statement.org/

Introduction: The manuscript provided a rationale for assessing epigenetic age acceleration (i.e. deviation of DNA methylation age from chronological age) among children with traumatic experiences. However, the theoretical explanation for assessing eye-tracking (gazing at eyes) as marker for social emotional problems is insufficient. This needs to be expanded, since it’s an independent effect from epigenetic aging. The only information provided was that eye gazing was a behavioral indicator of child maltreatment, but reader is left with uncertainty as to why that’s the case. For instance, children who are victims of maltreatment can suffer eye trauma, which can lead in itself to post-traumatic stress disorder (PTSD). PTSD can cause visual disturbances known as Post-Traumatic Vision Syndrome (PTVS). Alternatively, children who have experienced abuse may learn to avoid eye contact. Also, children may develop social anxiety or suffer from it concurrently. People with social anxiety may avoid eye contact because they perceive it as aversive. In times of extreme trauma, a child may suddenly stare into the distance rather than engage with others (issues in the social engagement system, which is also affected in children with autism, and related to polyvagal theory of trauma). Lastly, people with PTSD may have different pupil responses than those without PTSD. For example, their pupils may not constrict normally when presented with new visual stimuli, or they may dilate more in response to emotional stimuli. These aspects may be already known to the authors, but there are not readily evident to the reader. This reviewer is a pediatric neurologist and specializes in traumatic stress in children.

With regards to objectives, the manuscript presents a central hypothesis which was formally tested. Excellent description.

Methods: Design

As previously noted, the paper should clearly state the type of study design. If the data was collected at a single point in time (snapshot) from any given population (one sample or independent samples are still cross-sectional is the data collection is at single point in time), then the study is cross-sectional. The study would be a case control study if it’s retrospective, meaning the researchers look back in time to identify potential exposures that may have led to a disease. There is no cross-sectional case-control. Perhaps, this is a case control study that have measures collected cross-sectionally and some measures were retrospective. This reviewer does not have enough information to make this determination. If the authors took both epigenetic aging and eye gaze data concurrently but asked other questions about the past that serve as independent predictors, this needs to be clearly stated. Cross-sectional studies can ask questions about the past, but such questions are still collected at a single point in time (cross-sectional). Then, the data are cross-sectional, the study is cross-sectional. However, the authors must indicate which study design they consider in the methodology. Other infrequent study designs could also be nested case control studies if the case control is conducted as part of a prospective study (cohort or a clinical trial). Studies could also be case cohort studies, which are rarer design (In a case-cohort study, cases are defined as those participants of the cohort who developed the disease of interest, but controls are identified before the cases develop. This means that controls are randomly chosen from all cohort participants regardless of whether they have the disease of interest or not, and that baseline data can be collected early in the study). The following work is recommended: Janković, S. (2008). Observational Studies . In: Kirch, W. (eds) Encyclopedia of Public Health. Springer, Dordrecht. https://doi.org/10.1007/978-1-4020-5614-7_2378

The other important aspect is the primary or secondary use of data. If this is an analysis of existing data (regardless if the previous study was conducted by the same authors), then the study is a secondary data analysis. However, if the original study was ongoing and sustained a lapse in time, which was later compensated, then it’s still the same study. Just need to report it as primary data collection and explain the limitations of collecting data in two different times.

With regards to CM and TD group, it’s important to explain who children were recruited into the study. It’s great to know that study protocol was approved by ethics committee. However, there is a need to explain how subjects were protected from retraumatization, if any care was provided (e.g., counseling), and what were some rules to stop trial.

If this was a trial, what was the numbers needed to treat value, if any.

The researchers indicate that the original study did not have a sufficiently large sample size. However, the reader doesn’t have information in the current manuscript about power and sample size calculations and effect size for this study.

The Table 1 is results and should be presented in the Results section, not in the methods.

With regards to variables and measurement, it needs its own subheading in the methods so that the reader can make a distinction between dependent variables (e.g., epigenetic aging and eye gaze), independent variables (CM vs TD), and confounding (i.e., sociodemographics). The manuscript reports a series of instruments with their citation. This is good. However, variables in the study need to be first defined conceptually and then the measure is provided, with details on how the variables were operationalized in the study. The only two variables that were very carefully defined were the dependent variables “Calculation of epigenetic age acceleration” and “gaze patter measurement”. However, independent and confounding were not.

With regards to analysis, serial mediation analysis models were estimated with Lavaan (path analysis). Since the sample size is not very large either in the current study, it’s important to mention of there were adequate power and sample size calculations for the number of parameters in the path analysis. If the authors used latent variables, this needs to be reported. In this regard, the description needs to mention model statistics that were consulted (e.g., likelihood-ratio tests) and the goodness of fit indices that were assessed (e.g., Chi-square for the baseline model vs the estimated, Akaike, and RMSEA) to determine that the model was a good fit to the data.

There is no mention of how missing data was handled. This is very important, particularly for the path analysis (i.e. full information maximum likelihood vs listwise vs other).

Discussion:

The findings suggest an association between CM and the study dependent variables (epigenetic aging and gaze pattern). This association indicate a correlation, not causation, since the data are cross-sectional. Thus, the study needs to indicate this important limitation.

The issue of sample size considerations and power (or underpower) need to be discussed with regards to the results of the path analysis, which did not establish a direct sequential association

between mAge acceleration, visual attention to the eyes, and SDQ scores. Lack of significance may mean the test was underpowered. Power calculations for the path analysis were not provided, so the reader cannot identify if that’s the case.

This reviewer agrees that a study strength is that few studies have formally recognized cases of CM confirmed by child protective services.

With regards to study limitations, if the study is a cross-sectional study or case-control study, it will have different limitations if it’s one or the other. This reviewer is not familiar with a design cross-sectional case-control design. Please provide a citation and why this is the case.

I hope this comments are taken as constructive to improve the manuscript, since the hypothesis tested are very important and the information adds evidence to the body of knowledge albeit the limitations aforementioned.

6. PLOS authors have the option to publish the peer review history of their article (what does this mean? ). If published, this will include your full peer review and any attached files.

**Do you want your identity to be public for this peer review?** For information about this choice, including consent withdrawal, please see our Privacy Policy .

Reviewer #1: No

Reviewer #2: **Yes: ** Abraham Salinas-Miranda

---

## [Author Response · Author response to Decision Letter 1]

11 Feb 2025

Thank you for understanding and pointing out.I revised all.Reviewer #1: The manuscript by Ochiai et al. expanded on their initial pilot study of biological age in maltreated children by incorporating a larger sample size, which allowed them to do more thorough analyses. Importantly, they were able to replicate their previous findings, which concluded that biological age is accelerated in maltreated children compared to typically developing children. Although the major results presented here are not novel and have been previously reported elsewhere, I commend the authors for undertaking this study and replicating their prior work. I believe the following points should be addressed before this manuscript can be accepted for publication.

Thank you for understanding and appreciating that our study demonstrated that it was possible to replicate the results from our previous study with a larger sample size and that our more comprehensive analysis provides new discoveries. Below are our responses to your comments.

Major

1. I believe there is a typo in the title: “diffculties” should be “difficulties.”

We corrected the typo. Thank you for pointing this out.

2. I found the methods section, particularly the “Participants” subsection, a bit confusing. The authors mention on line 102 that the first group comprised 47 children, but the numbers of CM and TD in the parentheses do not add up. This also applies to the “remaining 65 children” mentioned in line 104. Could the authors please clarify?

Thank you for pointing this out. We have corrected this appropriately.

3. It appears from Table 1 that CM participants were statistically significantly older, chronologically, than the TD group used as controls. Given that chronological age is correlated with biological age, wouldn’t it be expected that older CM participants would exhibit higher biological age? Could the authors clarify how this was addressed in subsequent analyses?

As pointed out, the CM group was significantly older than the TD group, as illustrated in Table 1. However, we are comparing “mAge acceleration” and not mAge, which correlates with chronological age. However, as we are concerned about the effect of age difference, we have added chronological age as a confounding factor in our analysis. We have added a limitation regarding this point (P24, L481-493).

4. In the discussion (lines 355-357), the authors state that the number of maltreatment types/severity was not associated with mAge. Could they expand on this? Was this assessed in another path analysis, and can this data be shown?

Thank you for pointing this out. You are correct that an explanation of this analysis was missing. We have revised and reorganized the Statistical Analysis section (P17, LX320-371 to include a clear description of this analysis. Please note that this analysis is referred to as a sensitivity analysis, not a path analysis. The results of the sensitivity analysis are presented in Figure 4.

5. Given the mechanistic nature of this study, it would be beneficial if the authors further discussed the potential biological implications of mAge in the discussion (PMID: 30241605). Currently, they mention that epigenetic changes have been reported in various genes among CM participants and provide references. However, it would be useful if they discussed one or some of the genes/CpG regions assessed in their mAge model and perhaps speculate on some functional relevance to CM.

Thank you for the ideas to improve this. We have added the discussion the reviewer suggested (P20, L405-415).

Minor

1. In Table 1, I suggest providing values for each level of the categorical variables presented. For example, instead of providing data for only male participants, recode the variable as gender and provide information for all genders assessed in the study. Also, please clarify the parentheses for “ACE total.” The footnote for this table might benefit from a note clarifying that CM participants can be polyvictimized, which is why the numbers do not add up to the sample size. Additionally, consider removing the row with information about sexual abuse since it is entirely NA.

As suggested, we have corrected Table 1.

2. Consider showing individual data points for Figure 1 and adding p-values to all figures for consistency.

There was some error, and the data point in Figure 1 was missing. We have replaced it with the correct one. Thank you for pointing this out.

3. For Figure 3, consider showing additional trend lines—one specific for CM and one for TD.

As suggested, we have added the trend lines: one specifically for CM and one for TD.

4. It might benefit readers if the authors referenced specific figures in the discussion. Given some mixed results between path analysis and simple correlations (e.g., mAge correlating with gaze pattern), this could provide clearer context for readers.

As suggested, we have referenced specific figures in the discussion to make this more reader friendly.

Reviewer #2: I was asked to review the article entitled “Behavioral and emotional difficulties in maltreated children: Associations with epigenetic clock changes and visual attention to social cues”, submitted to PLOS ONE. The article examines an interesting topic, which is accelerated aging among individuals exposed to child maltreatment. There is currently a research gap (i.e., evidence about telomere length is inconsistent). However, there are some issues that need to be revised before this article can be accepted for publication. Details of my assessment is below.

We appreciate your detailed comments and suggestions on how to strengthen our manuscript. Below are our responses to the comments.

There is a missing “i” in the word difficulties in the title.

Title: Behavioral and emotional diffculties in maltreated children: Associations with epigenetic clock changes and visual attention to social cues

We have corrected the typo. Thank you for pointing this out.

Abstract: The study design is missing in the abstract and title. This must be explicit in the title or abstract for bibliographic searches and systematic reviews. It’s recommended the authors examine the STROBE guidelines for observational studies https://www.strobe-statement.org/

Thank you for the suggestion. We put “case-control design” in the appropriate location in the abstract (P2, L27).

Introduction: The manuscript provided a rationale for assessing epigenetic age acceleration (i.e. deviation of DNA methylation age from chronological age) among children with traumatic experiences. However, the theoretical explanation for assessing eye-tracking (gazing at eyes) as marker for social emotional problems is insufficient. This needs to be expanded, since it’s an independent effect from epigenetic aging. The only information provided was that eye gazing was a behavioral indicator of child maltreatment, but reader is left with uncertainty as to why that’s the case. For instance, children who are victims of maltreatment can suffer eye trauma, which can lead in itself to post-traumatic stress disorder (PTSD). PTSD can cause visual disturbances known as Post-Traumatic Vision Syndrome (PTVS). Alternatively, children who have experienced abuse may learn to avoid eye contact. Also, children may develop social anxiety or suffer from it concurrently. People with social anxiety may avoid eye contact because they perceive it as aversive. In times of extreme trauma, a child may suddenly stare into the distance rather than engage with others (issues in the social engagement system, which is also affected in children with autism, and related to polyvagal theory of trauma). Lastly, people with PTSD may have different pupil responses than those without PTSD. For example, their pupils may not constrict normally when presented with new visual stimuli, or they may dilate more in response to emotional stimuli. These aspects may be already known to the authors, but there are not readily evident to the reader. This reviewer is a pediatric neurologist and specializes in traumatic stress in children.

Thank you for the detailed explanation and the suggestions. You were right that we should add these, so we have revised the manuscript (P5, L91-111).

With regards to objectives, the manuscript presents a central hypothesis which was formally tested. Excellent description.

Thank you for the compliments; we appreciate it.

Methods: Design

As previously noted, the paper should clearly state the type of study design. If the data was collected at a single point in time (snapshot) from any given population (one sample or independent samples are still cross-sectional is the data collection is at single point in time), then the study is cross-sectional. The study would be a case control study if it’s retrospective, meaning the researchers look back in time to identify potential exposures that may have led to a disease. There is no cross-sectional case-control. Perhaps, this is a case control study that have measures collected cross-sectionally and some measures were retrospective. This reviewer does not have enough information to make this determination. If the authors took both epigenetic aging and eye gaze data concurrently but asked other questions about the past that serve as independent predictors, this needs to be clearly stated. Cross-sectional studies can ask questions about the past, but such questions are still collected at a single point in time (cross-sectional). Then, the data are cross-sectional, the study is cross-sectional. However, the authors must indicate which study design they consider in the methodology. Other infrequent study designs could also be nested case control studies if the case control is conducted as part of a prospective study (cohort or a clinical trial). Studies could also be case cohort studies, which are rarer design (In a case-cohort study, cases are defined as those participants of the cohort who developed the disease of interest, but controls are identified before the cases develop. This means that controls are randomly chosen from all cohort participants regardless of whether they have the disease of interest or not, and that baseline data can be collected early in the study). The following work is recommended: Janković, S. (2008). Observational Studies . In: Kirch, W. (eds) Encyclopedia of Public Health. Springer, Dordrecht. https://doi.org/10.1007/978-1-4020-5614-7_2378

Thank you for defining the study design and providing examples of it. Indeed, we had written “cross-sectional case-control design,” which was our mistake. This study was a case-control study, as described in the abstract in this revised version. We learned a lot from reading the STROBE guidelines you provided. Thank you for your suggestions.

The other important aspect is the primary or secondary use of data. If this is an analysis of existing data (regardless if the previous study was conducted by the same authors), then the study is a secondary data analysis. However, if the original study was ongoing and sustained a lapse in time, which was later compensated, then it’s still the same study. Just need to report it as primary data collection and explain the limitations of collecting data in two different times.

As suggested, we have reported that the first half of the population was used as for the secondary data analysis (P7, L134-136).

With regards to CM and TD group, it’s important to explain who children were recruited into the study. It’s great to know that study protocol was approved by ethics committee. However, there is a need to explain how subjects were protected from retraumatization, if any care was provided (e.g., counseling), and what were some rules to stop trial.

We appreciate the reviewer’s concern regarding potential retraumatization during the study. To clarify, the experimental content of this study was explicitly designed to avoid causing retraumatization. Questionnaires addressing childhood adversities, such as ACE, were not administered directly to the participants but were instead completed by their caregivers (not always their biological parents). Consequently, there were no counseling sessions or predefined protocols for halting the examination in such cases, as no such situations were anticipated. Nonetheless, we ensured that general ethical considerations were thoroughly addressed throughout the study design.

If this was a trial, what was the numbers needed to treat value, if any.

This was not a clinical trial. We answer the question about power analysis in the next section.

The researchers indicate that the original study did not have a sufficiently large sample size. However, the reader doesn’t have information in the current manuscript about power and sample size calculations and effect size for this study.

As suggested, we have added the power and sample size calculations in the Methods section (P11, L211–217) and the effect size in the Results section (P19, L365-371).

The Table 1 is results and should be presented in the Results section, not in the methods.

As suggested, we have moved Table 1 to the Results section.

With regards to variables and measurement, it needs its own subheading in the methods so that the reader can make a distinction between dependent variables (e.g., epigenetic aging and eye gaze), independent variables (CM vs TD), and confounding (i.e., sociodemographics). The manuscript reports a series of instruments with their citation. This is good. However, variables in the study need to be first defined conceptually and then the measure is provided, with details on how the variables were operationalized in the study. The only two variables that were very carefully defined were the dependent variables “Calculation of epigenetic age acceleration” and “gaze patter measurement”. However, independent and confounding were not.

Thank you for your advice. As suggested, we have added subheadings to explain each analysis’s variables and measurements (P11, L224-251).

With regards to analysis, serial mediation analysis models were estimated with Lavaan (path analysis). Since the sample size is not very large either in the current study, it’s important to mention of there were adequate power and sample size calculations for the number of parameters in the path analysis. If the authors used latent variables, this needs to be reported. In this regard, the description needs to mention model statistics that were consulted (e.g., likelihood-ratio tests) and the goodness of fit indices that were assessed (e.g., Chi-square for the baseline model vs the estimated, Akaike, and RMSEA) to determine that the model was a good fit to the data.

Thank you for your insightful comments regarding the serial mediation analysis. Regarding the use of latent variables, the model did not include any latent variables, as all variables in the analysis were observed variables. As you noted, due to the small sample size required to perform SEM, we modeled it with lavaan using the bootstrap method. The bootstrap method offers several benefits in SEM, particularly when sample sizes are small. It improves the estimation of standard errors by deriving them directly from the sampling distribution without relying on assumptions like normality. This makes it effective for handling non-normal or skewed data. Additionally, the bootstrap method provides reliable confidence intervals for path coefficients and indirect effects, which are often challenging to estimate with traditional methods. By stabilizing parameter estimation, the bootstrap method is especially useful in studies with limited sample sizes, enhancing the overall reliability of the analysis. We have added these points to the Methods (P13, L267–269) and Discussion section (P24, L490–493).

Next, the initial path model was a saturated model with zero degrees of freedom, which made it impossible to calculate model fit indices. Saturated models always perfectly reproduce the observed covariance matrix; thus, model fit indices are not meaningful in such cases. To address this limitation, we revisited the model and removed the non-significant path d12 (the path from mAge acceleration to visual attention to the eyes), which allowed us to introduce degrees of freedom. We then reanalyzed the model, and the resul

---

## [Editor Report · Decision Letter 1]

14 Mar 2025

Behavioral and emotional diffculties in maltreated children: Associations with epigenetic clock changes and visual attention to social cues

PONE-D-24-33599R1

Dear Dr. Tomoda,

We’re pleased to inform you that your manuscript has been judged scientifically suitable for publication and will be formally accepted for publication once it meets all outstanding technical requirements.

Kind regards,

Abraham Salinas-Miranda, MD, PhD

Academic Editor

PLOS ONE

Additional Editor Comments (optional):

The authors have addressed point by point the requested revisions from reviewers. The recommendation is to accept.
---

## [Editor Report · Acceptance letter]

PONE-D-24-33599R1

PLOS ONE

Dear Dr. Tomoda,

I'm pleased to inform you that your manuscript has been deemed suitable for publication in PLOS ONE. Congratulations! Your manuscript is now being handed over to our production team.

Kind regards,

on behalf of

Dr. Abraham Salinas-Miranda

Academic Editor

PLOS ONE